



# A pyroelectric thermal sensor for automated ice nucleation detection

Fred Cook[1], Rachel Lord[2], Gary Sitbon[1], Adam Stephens[1], Alison Rust[2], and Walther Schwarzacher[1]

[1]H. H. Wills Physics Laboratory, University of Bristol, Tyndall Avenue, Bristol BS8 1TL, United Kingdom
[2]School of Earth Sciences, University of Bristol, Wills Memorial Building, Queens Road, Bristol BS8 1RJ, United Kingdom

**Correspondence:** Fred Cook (fred.cook@bristol.ac.uk)

**Abstract.**

A new approach to automating droplet freezing assays is demonstrated by comparing the ice nucleating efficiency of a K-feldspar glass and a crystal with the same bulk composition. The method uses a pyroelectric polymer PVDF (polyvinylidene fluoride) as a thermal sensor. PVDF is highly sensitive, cheap and readily available in a variety of sizes. As a droplet freezes
latent heat is released, which is detected by the sensor. Each event is correlated with the temperature at which it occurred. The sensor has been used to detect microlitre volume droplets of water freezing, from which frozen proportion curves and nucleation rates can be quickly and automatically calculated. Our method shows glassy K-feldspar to be a poor nucleator compared to the crystalline form.

## 1 Introduction

Ice nucleation is of great importance, particularly to atmospheric science, where the presence of ice nucleating particles (INPs) can drastically change the temperature at which supercooled droplets of water freeze. This in turn has a large impact on the lifetime, precipitation and other important properties of clouds(Murray et al., 2012; Hoose and Möhler, 2012). Accurate cloud modelling faces several barriers, since atmospheric processes and the interactions of droplets within clouds are complex, e.g. the Bergeron–Findeisen process (Pruppacher and Klett, 1997), and capturing them with available computing power is not a
straightforward task. However, more fundamentally, the kinetics behind the different modes of heterogeneous ice nucleation (immersion, deposition, condensation and contact) on INPs are not well understood.

It is assumed that each INP has preferential areas for ice nucleation, active sites, the exact arrangement and nucleating ability of which are unique to any individual INP (Holden et al., 2019). Direct investigation into the formation of ice at these active sites is difficult due to the stochastic nature of nucleation and the small size (nanometre scale) of the initial ice nucleus.
Although computational modelling provides insight into the favoured structures of water molecules as they freeze on surfaces there are still many limitations, mostly due to the time-scale problem (Sosso et al., 2016). At all but the lowest temperatures spontaneous nucleation events are very rare. To capture them in simulations requires a compromise between the accuracy of water molecule model, number of water molecules in the system and total simulation time. Coarse grained water models can simulate on the order of $10^6$ molecules for around one millisecond(English and Tse, 2015), more detailed models reduce the
number of molecules to $10^5$ on a similar time scale, and ab initio calculations are currently limited to around 100 molecules. These numbers may be compared to a picolitre of water, at the smaller end of the experimental scale, which contains on





the order of $10^{13}$ molecules and can remain liquid for hours even at very low temperatures. One way to reduce the time necessary is by careful seeding of molecules into ice-like structures, however, this can lead to unpredictable biases in the results. Experimentally the time-scale problem is not an issue, as experiments can last for days if necessary(Heneghan and

Haymet, 2003) and larger volumes of water can be used to greatly increase the chance of a nucleation event being observed.

There are many experimental methods for determining nucleation rates, including levitators(Jing et al., 2019; Krämer et al., 1999; Lü and Wei, 2006), cloud chambers(Möhler et al., 2003), continuous flow diffusion chambers (CFDCs) (Rogers, 1988; Kanji and Abbatt, 2009; Hiranuma et al., 2015; Chou et al., 2011; Stetzer et al., 2008) and cold plate droplet arrays (Hiranuma et al., 2015; Häusler et al., 2018; Gibbs et al., 2015; Campbell et al., 2015; Whale et al., 2015; Tobo, 2016; Tarn

et al., 2018), each able to probe different conditions for nucleation(Demott et al., 2018). For instance CFDCs allow control of the vapour saturation over ice, enabling deposition and immersion mode nucleation to be investigated. However, assumptions have to be made about the mode of nucleation according to the relative humidity, with deposition mode or pore condensation mode(Marcolli, 2014)assumed below 100% and immersion/condensation mode assumed above(Boose et al., 2019). Furthermore, there is an upper temperature limit, suggested by Hiranuma et al.(Hiranuma et al., 2015) to be -9 °C beyond which the

saturation conditions cannot be maintained and there is also the issue of particle detection, e.g. Tobo et al.(Tobo et al., 2013) were unable to detect particles smaller than 0.5 μm. Cloud chambers are an attractive alternative for atmospheric scientists as they recreate the natural dynamics of cloud formation over a wide range of temperatures. However, they also suffer from problems with detection of small particles, as well as particles settling out in the course of the experiment, leading to biases in the ice nucleation rates obtained(DeMott and Rogers, 1990). A problem common to both CFDCs and cloud chambers is that

they can only probe small numbers of particles, which makes evaluation of poor INPs difficult, as nucleation events are rare.

For studying immersion mode ice nucleation, cold plate arrays are especially useful. A typical cold plate array is shown in Figure 1. Most immersion mode droplet array ice nucleation experiments use droplets on the order of picolitres to microlitres. In general this method involves pipetting an array of droplets onto a cold plate, although microfluidic generators(Tarn et al., 2018) and droplet printers(Peckhaus et al., 2016) are also used. The droplets are then cooled, usually with a linear decrease

in temperature, although temperature steps are also used(Gibbs et al., 2015), with the freezing temperature of each droplet recorded. The frozen fraction is measured as a function of temperature, from which a nucleation rate can be calculated(Whale et al., 2017). By using a cold plate droplet array the effects of varying INP concentrations over several orders of magnitude can be investigated. As only one nucleation event is required to freeze a droplet, even the nucleating ability of poor INPs can be tested. Of course cold plate arrays also have drawbacks. For example, since the droplets sit on a substrate, it is essential to

exclude substrate-induced nucleation. It also essential to control the purity of the water used to form the droplets as even traces of contaminant could affect the nucleation probability.

Without automation, determining the temperature at which each droplet freezes is a time consuming process, especially for the large number of droplets required to compensate for the stochastic nature of nucleation. Freezing events are usually detected via a change in the optical properties such as a change in transparency, or via the latent heat released. Optical detection

has been automated(Peckhaus et al., 2016; Budke and Koop, 2015; Stopelli et al., 2014; Reicher et al., 2018), with software to recognise the locations of droplets and monitor the associated pixel intensity, which goes through a sudden change at the point





of freezing. This effect can be enhanced using polarizers to take advantage of the birefringence of ice(Peckhaus et al., 2016). However, automation is not completely straight forward, as it requires large amounts of data processing and storage to analyse images of the droplets, as well as ways to avoid artefacts leading to false identification of freezing events. For instance, droplets

can move during cooling, which can lead to a change in measured pixel intensity unless each droplet is tracked, and movement in the lab can lead to shadows or reflections over the droplet, also causing a possible change in measured pixel intensity.

The latent heat of crystallization can be detected by monitoring the infrared emissions of droplets(Zaragotas et al., 2016; Harrison et al., 2018; Kunert et al., 2018), or via calorimetry. Differential scanning calorimetry (DSC) has been widely used(Riechers et al., 2013; Parody-Morreale et al., 1986; Yao et al., 2017; Kaufmann et al., 2017; Kumar et al., 2018) to

study ice nucleation. However DSC is not directly comparable to other methods discussed here as it cannot detect individual droplets freezing. Infrared thermometry(Harrison et al., 2018) has the advantage that it can also be used to measure the temperature of droplets as they freeze, revealing any thermal gradients across the set-up which may otherwise be neglected. However, due to the Stefan-Boltzmann law infrared thermometry at low temperatures is limited to large droplets.

The latent heat can also be detected by other kinds of thermal sensor. Here we present a particularly simple, cheap and adapt-

able pyroelectric polymer based device for this purpose. The pyroelectric polymer used is polyvinylidene fluoride (PVDF), which can be bought in large pre-metallised sheets and cut to shape. This adaptability means it can be incorporated into many standard droplet array experiments. The latent heat released by droplets provides a clear and unambiguous signal which can be easily converted to a list of droplet freezing temperatures for further analysis. We provide information on how our PVDF sensor was optimized, as well as details of the associated charge amplifying circuitry.

To demonstrate the effectiveness of our sensor, we present data comparing the nucleating ability of a standard sample of crystalline K-feldspar (BCS-CRM 376/1, as used by Atkinson et al.(Atkinson et al., 2013)) with a glassy sample having the same bulk chemical composition. K-feldspar has been shown to be an important contributor to the ice nucleation activity of mineral dust aerosol(Atkinson et al., 2013; Yakobi-Hancock et al., 2013) and has therefore been studied extensively(Kiselev et al., 2017; Whale et al., 2017; Zolles et al., 2015; Pedevilla et al., 2016; Peckhaus et al., 2016; Harrison et al., 2016; Augustin-

Bauditz et al., 2014; Kumar et al., 2018). For example Kiselev et al.(Kiselev et al., 2017) showed that, at least in deposition mode, ice preferentially forms on the high energy (100) surface, only exposed in cracks and defects, not on the most easily cleaved (001) surface as previously suggested(Pedevilla et al., 2016). Despite the insight this provides into the nature of active sites, there is no guarantee that the same applies to immersion mode freezing. Indeed, recent molecular dynamic simulations by Soni and Patey (Soni and Patey, 2019) of water molecules on clean (001), (010) and (100) surfaces of microcline K-feldspar

show no evidence of ice nucleation, further suggesting the importance of defects in ice nucleation. In order to investigate the importance of the presence of crystalline surfaces at active sites a standard crystalline sample of K-feldspar is compared to a glassy sample of the same bulk composition.

Our glassy sample was made by melting, quenching and grinding the crystalline sample. Quenching the sample means the long range order of a crystalline structure is not given time to form, leading to an amorphous structure more similar to that of the

liquid form. A similar approach was recently used by Maters et al. (Maters et al., 2019) in comparing natural crystalline samples and their glassy equivalents. The difference in local structure alone could lead to the glassy and crystalline samples having very



different ice nucleation behaviours. However, it is also necessary to consider their different mechanical properties(Debenedetti, 1996). Crystals can be cleaved along preferred surfaces, often resulting in flat faces, although there will also be a number of defects present. Glasses do not have long range order, leading to irregular shapes when they are mechanically ground, with very

different surface structure to the crystalline form. Surface topography has been shown to be extremely important in determining ice nucleating efficiency (Holden et al., 2019; Whale et al., 2017). In addition, the interaction of water and INP is complex, and the chemical nature of bonds at the surface as well as the structure play an important and interconnected role. Even if crystalline and glassy samples have the same bulk chemical composition, their surface chemistries could differ.

The difference in ice nucleating efficiency between crystalline and glassy samples is of considerable practical importance, as

material glassy samples are not just of interest for their different structural properties. Particles dispersed by volcanic eruptions include a mixture of glassy and crystalline aluminosilicates, with the proportions of components varying widely between eruptions(Wright et al., 2012; Cashman and Rust, 2016). The ice nucleating ability of particles within the plume is of great interest since the prevalence and effectiveness of INPs within a plume will have a large effect on its lifetime and dynamics, knowledge of which is vital for accurate forecasting(Macedonio et al., 2016).

## 2 Thermal Sensor Design

A pyroelectric material has a temperature dependent spontaneous electric polarisation (Whatmore, 1986). As the temperature of the pyroelectric element changes the spontaneous polarisation also changes, causing a build-up of charge at the surface. Unlike the thermoelectric effect, a temperature gradient is not required, just an absolute temperature change. If the surfaces are metallised the pyroelectric element can be thought of as a parallel plate capacitor which is charged by changes in temperature.

Not all PVDF is pyroelectric; it must be mechanically stretched in the presence of a strong electric field to induce a spontaneous net dipole moment. The PVDF used here was purchased from *Piezotech*, pre-stretched and metallised with approximately 200 nm gold on top of 50 nm chromium on both sides. Three different thicknesses, 9 µm, 52 µm and 110 µm were purchased. The as-delivered 10 cm × 10 cm sheets were cut to shape, in our case circles 20 mm in diameter to sit on the silver cooling block of a Linkam THMS600 cooling stage, shown in Figure 2a. When cutting it is easy to crimp the two surfaces together

accidentally, electrically shorting the two sides meaning no charge will be measured. Such short circuits can be detected by testing for continuity with a multimeter.

In use the PVDF is held against the cooling block using a custom machined plastic (PTFE) clamp. This grips the edge of the cooling block and is pushed down to exert a small amount of pressure on the PVDF to keep it flat, as well as to hold a wire in contact with the upper surface. Contact with the lower electrode is made via the cooling block, which is grounded. Before an

experiment, the top gold surface supporting the droplet array is coated with Vaseline to make it hydrophobic (Tobo, 2016).

When using pyroelectric materials both the thermal and electrical properties of the system must be considered. Since the response from the PVDF depends on the absolute temperature change a thermally isolated pyroelectric element with as small a thermal mass as possible will give the greatest signal for any given input. However, the requirement for thermal isolation conflicts with the requirement for excellent thermal conductivity to keep the droplets in thermal equilibrium with the cooling



block. In practice even the thickest PVDF had sufficient thermal conductivity to maintain equilibrium with the cooling block at a cooling rate of $1\,^{\circ}\mathrm{C\,min}^{-1}$, and low enough thermal mass that the temperature rise associated with the latent heat released by the freezing of a single microlitre droplet can be detected reproducibly.

   The thickness of the PVDF also dictates its capacitance, which will have an effect on the electrical circuit used to detect the voltage change resulting from any temperature change. We constructed a charge amplifier using an LT1793 low noise

operational amplifier, in conjunction with a feedback capacitor, $C_f$, and feedback resistor, $R_f$ as shown in Figure (2b). In the absence of the feedback resistor the feedback capacitor would be saturated by the charge that the PVDF releases as the temperature of the stage is lowered, even before any droplets froze. Using the feedback resistor there is a small negative offset to the signal output from the charge amplifier, proportional to the cooling rate. When a droplet freezes the temperature of the PVDF increases rapidly and transiently, due to the latent heat released. The pyroelectric effect produces a charge on the

metallised surfaces of the PVDF that charges $C_f$ and therefore gives rise to a positive spike in the output signal. The spikes decay exponentially with the characteristic electrical time constant of the circuit ($\approx 20\,\mathrm{ms}$). The output from the charge amplifier was monitored using an analogue to digital converter (NI USB-6002), sampled at $1\,\mathrm{kHz}$, which is fast enough to detect all droplets freezing, without creating unnecessarily large data files. Data acquisition was controlled collected by a LabVIEW program, which also controlled the temperature of the cooling stage. The LabVIEW program returned an array with three columns; time,

cooling block temperature and sensor output signal.

   Figure 3 shows a comparison of the voltage responses of the three different thicknesses of PVDF available when microlitre droplets of pure water freeze. The RMS noise values were computed for each thickness between 0 and -5 $^{\circ}$C, before any droplets had frozen. These were 0.096 V, 0.1 V and 0.104 V for the $9\,\mu\mathrm{m}$, $52\,\mu\mathrm{m}$ and $110\,\mu\mathrm{m}$ samples respectively. The small increase in noise with thickness is due to the fact that all pyroelectric materials are also piezoelectric. Any mechanical vibra-

tions, for instance due to liquid nitrogen being pumped through the stage, will produce a signal proportional to the amount of piezoelectric material present. Other than this, the noise value for each foil thickness is equivalent to within a few percent, consisting of a slow random oscillation superimposed with a $50\,\mathrm{Hz}$ oscillation due to mains interference, despite shielding of both the PVDF element and charge amplifying circuitry. Figure 3 shows that the average peak height is inversely related to the thickness of the PVDF used. The average peak height to RMS noise ratios are $5.1 \pm 0.6$, $2.3 \pm 0.4$ and $1.4 \pm 0.2$ for the

$9\,\mu\mathrm{m}$, $52\,\mu\mathrm{m}$ and $110\,\mu\mathrm{m}$ samples respectively. All of these values were found using a $57\,\mathrm{pF}$ feedback capacitor in parallel with a $10\,\mathrm{M\Omega}$ feedback resistor. The low thermal mass of the thinnest sample of PVDF leads to the highest absolute temperature change from the latent heat released, and therefore the largest signal.

   In principle, the area under the peak corresponding to a droplet freezing is proportional to the latent heat released, and PVDF foils have previously been used as calorimeters (Etzel et al., 2010; Lew et al., 2010; Coufal and Hefferle, 1985).

However, this isn't possible in the present experimental arrangement for two reasons. Firstly, the situation is complicated by the continuously decreasing temperature of the cooling block, requiring the feedback resistor. Secondly, PVDF has large variations in pyroelectric constant across the surface(Lang and Das-Gupta, 1984) because during the poling process the PVDF is typically stretched up to four times its original length, leading to macroscopic crystalline and amorphous regions. Hence there is a large spatial variation in pyroelectric response. The variation in pyroelectric response means that the output signal for





the same release of latent heat also varies. This can be seen in Figure 3, where the spike heights have considerable variation for each thickness, despite the droplets being nominally the same size (errors are discussed in the results section). Hence the voltage data cannot be used to quantify the energy released by a droplet freezing, only to show that a freezing event occurred. An alternative pyroelectric material is lithium tantalate ($LiTaO_3$), as used by Frittman et al.(Frittmann et al., 2015) As it is a single crystal the spontaneous polarisation is much more uniform spatially, however, this also makes it much more fragile and

less adaptable to experimental set-ups than PVDF.

## 2.1 Sample Preparation

The crystalline K-feldspar comes from the Bureau of Analysed Samples (BCS-CRM No. 376/1), as used by Atkinson et al.(Atkinson et al., 2013) No further processing of this sample was done. The glassy sample was made by melting and rapid quenching the crystalline powder, followed by milling.

A range of mass fraction suspensions were made up gravimetrically for each sample, using Milli-Q $18.2\,M\Omega$ water. All experiments were completed within a week of the suspensions being made. Before pipetting onto the cold stage each sample was ultrasonicated for 15 minutes to break up aggregates. Samples were kept in sealed glass vials which were previously cleaned by filling the vials with nitric and sulphuric acid for 30 minutes each, before thorough rinsing with Milli-Q $18.2\,M\Omega$ water. They were stored out of direct light.

## 3   Results and Discussion

The surface area of both samples was measured via Brunauer-Emmett-Teller (BET) nitrogen gas absorption. Three repeats were taken, with the mean to extreme range used as the error. The values were $5.0 \pm 0.7\,\mathrm{m^2g^{-1}}$ and $1.8 \pm 0.4\,\mathrm{m^2g^{-1}}$ for crystalline and glassy K-feldspar respectively(†ESI). The percentage errors associated with the surface area per unit mass dominate the error in calculating surface area present in each droplet, but are comparable to other experiments. There are also

errors associated with the volume of each droplet pipetted, and amount of material which settled out of suspension during pipetting (Tarn et al., 2018), however, these are insignificant in comparison.

   A typical voltage-time graph is shown in Figure 4. The difference in peak height despite all of the droplets being the same size to the precision of the pipette ($\pm 0.03\,\mu l$) is visible, for the reasons discussed in section 2. Each assay of droplets produced a similar graph, which was converted to a list of freezing temperatures using a Python script. The thermocouple built into

the liquid nitrogen cooled stage was used to measure the temperature, which was observed to oscillate by $\pm 0.2\,°C$ due to small fluctuations in the pumping rate. On top of this there was an unknown thermal lag due to the PVDF and the Vaseline on which the droplets were placed. This was estimated to be a maximum of approximately $+0.8\,°C$, based on literature values for the thermal conductivity of PVDF, leading to the asymmetric error bars shown in Figure 5. The freezing of pure water (Milli-Q $18.2\,M\Omega$) starts at higher temperatures than we would expect from the homogeneous parameterisation by Atkinson

et al.(Atkinson et al., 2016) This was also noted by Whale et al.(Whale et al., 2015) and attributed to the greater chance of contamination due to the large droplet size, although the source was unknown. As Tobo(Tobo, 2016) reached the homogeneous





limit with microlitre droplets on Vaseline using a clean bench we assume that the source of the contamination is airborne(Polen et al., 2018).

Frozen fraction curves for the different mass fractions of glassy and crystalline K-Feldspar studied are shown in Figure 5a, along with the background freezing rate of the instrument. The dashed lines are generalised logistic functions (see supplementary information). These have no basis in theory, but provide good, monotonically decreasing, lines of best fit which can be differentiated analytically. From these curves the ice nucleation active site density, $n_s$, and the heterogeneous nucleation rate, $j_{\text{het}}$, were calculated. Equation 1(Connolly et al., 2009) was used to determine $n_s$

$$\frac{N - n(T)}{N} = 1 - \exp[-n_s(T)s], \tag{1}$$

where $n(T)$ is the number of liquid droplets out of a total population $N$ at temperature $T$ and $s$ is the surface area of INP per droplet. Values for $n_s$ for each concentration are shown in Figure 5b.

The results can also be interpreted in terms of a heterogeneous nucleation coefficient, $j_{\text{het}}$, normalised by the surface area of INP present. A population of $n$ liquid droplets containing an INP surface area $s$ per droplet will freeze over time as shown in equation 2,

$$\frac{\mathrm{d}n}{\mathrm{d}t} = -j_{\text{het}}(T)sn. \tag{2}$$

By applying the chain rule equation 3 is obtained,

$$-j_{\text{het}}(T) = \frac{\mathrm{d}n}{\mathrm{d}T}\frac{\mathrm{d}T}{\mathrm{d}t}\frac{1}{sn_l(T)}, \tag{3}$$

where $\mathrm{d}T/\mathrm{d}t$ is a constant cooling rate, $-1/60\,^{\circ}\mathrm{Cs}^{-1}$ for all experiments here. The individual data points in Figure 5c are from a numerical differentiation of the frozen fraction curves in Figure 5a using a second order central difference method. The

dashed lines are from an analytical differentiation of the fits to the frozen fraction curves (see supplementary information).

The calculation of $j_{\text{het}}$ from fraction frozen data is least reliable at the lowest temperatures. At lower temperatures there are few liquid droplets remaining, leading to a break down in the approximation that equations 1 and 3 are based on, that $\Delta n/n$ remains small(Koop et al., 1997). Also, as the temperature falls the probability that there would be multiple nucleation events in a single droplet increases(Atkinson et al., 2016). These factors lead to greatly increased errors in the nucleation rate calculated

at low temperatures. There is also an effect from the fact that our droplets are not perfectly uniform, due to variations in the amount of nucleant present, and the effectiveness of nucleant in any given droplet. The value of $j_{\text{het}}(T)$ found for glassy and crystalline K-feldspar here represents the average particle. As discussed by Kubota (Kubota, 2019) those droplets which are below the average $j_{\text{het}}$ are more likely to survive to lower temperatures, leading to a reduction in the measured nucleation rate.

Although the 1%wt suspensions of glassy K-feldspar showed some nucleating ability at higher temperatures, the gradient

of the frozen fraction curve remained much shallower than the crystalline form. While the nucleation active site density for

crystalline K-feldspar was similar to that measured by Whale et al.(Whale et al., 2015) using microlitre volume droplets, the active site density of glassy K-feldspar is approximately two orders of magnitude less at -20 °C. The heterogeneous nucleation rates also show clear separation between the glassy and crystalline phase. However, further experiment is needed to determine whether the importance of the crystalline form derives from its atomic order, its surface chemistry or its microstructure. For

example, a crystal can have well defined steps and terraces at the surface, which are absent in a glass. The greatly reduced nucleating ability suggests the importance of the presence of the crystalline form at whichever active sites are responsible for the nucleating effectiveness of K-feldspar.

## 4  Conclusions

We have shown that the pyroelectric thermal sensor is effective in gathering ice nucleation data. The sensor produces an

unambiguous signal for each microlitre droplet freezing event. Once a freezing run is finished the collected data can be passed into a Python script to extract a list of freezing temperatures. The script only takes a few seconds to run, and the data does not require any pre-treatment, greatly reducing the total time for experiments. The method is also easily adaptable to fit a wide range of cold plate arrays, allowing faster throughput for many experiments. Alternative pyroelectric materials such as Lithium Tantalate ($LaTiO_3$) could deliver improved performance, including the ability to quantify the heat released on freezing,

though at the cost of being more fragile. The effectiveness of the sensor has been demonstrated with an experiment comparing crystalline and glassy K-feldspar, with the results strongly suggesting the importance of crystalline structure in the nucleating ability of K-feldspar.

*Code and data availability.*  All data and code are available on request.

*Author contributions.*  WS and AR devised the project. FC and AS developed the thermal sensor. FC gathered the nucleation data and wrote

the LabVIEW and Python code for analysis. RL performed the BET analysis. Samples were provided by AR. GS provided lab assistance. FC wrote the paper, with input from WS and AR

*Competing interests.*  The authors declare that they have no conflict of interest.

*Acknowledgements.*  This work was funded by a Leverhulme Trust Research Project Grant, no. RPG-2014-180 and by EPSRC through their GCRF Institutional Sponsorship.





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





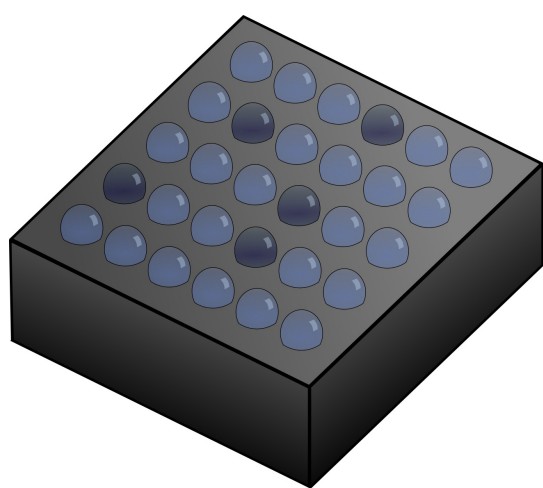

**Figure 1.** Schematic of a typical cold plate array, with droplets arranged in a grid on a heat sink. The heat sink is typically cooled by liquid Nitrogen or a Peltier device. The diagram shows some droplets frozen (dark).

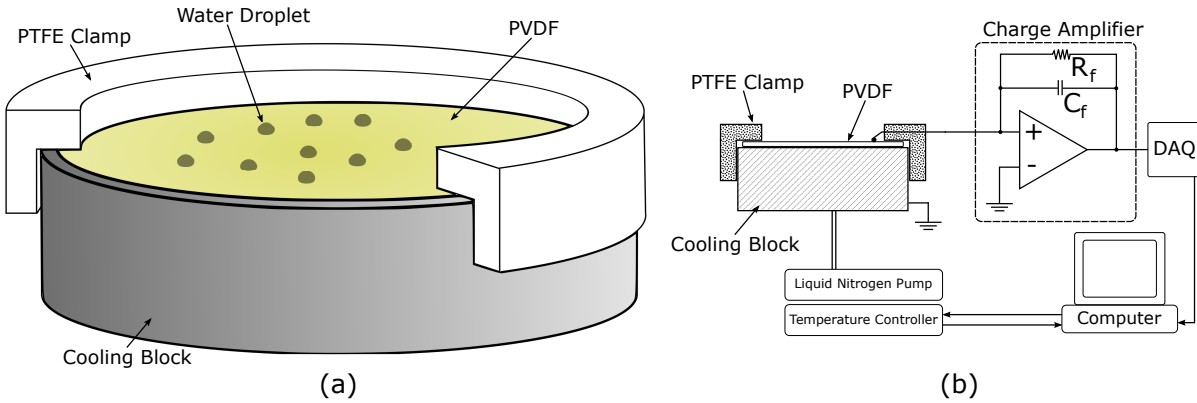

**Figure 2.** (a) A cut-away diagram of the cooling block with PVDF and clamp in place. The wire used to make contact with the upper surface is not shown. (b) A schematic of the experimental set-up including a simplified circuit diagram of the charge amplifier.



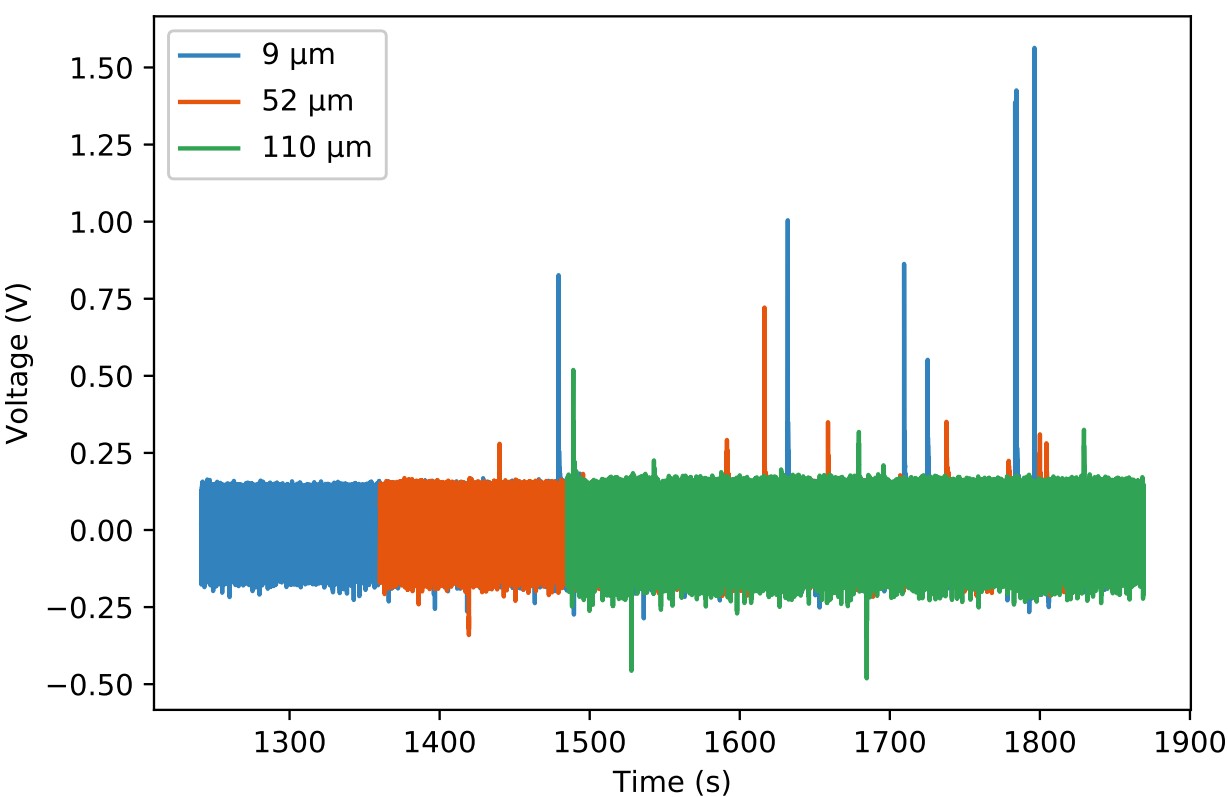

**Figure 3.** Sections of voltage time graphs for three different thicknesses of PVDF overlaid on top of each other. Each positive spike represents the freezing of a microlitre droplet of pure water. The offset at the start shows the similar noise amplitude for each thickness.



**Figure 4.** Raw data from a typical experimental run, in this case pure water droplets measured to determine the background freezing rate of the instrument. Each spike represents a droplet freezing, as shown in the upper graph and corresponding pictures. Approximate temperatures corresponding to the start and end of the run are shown at the bottom.



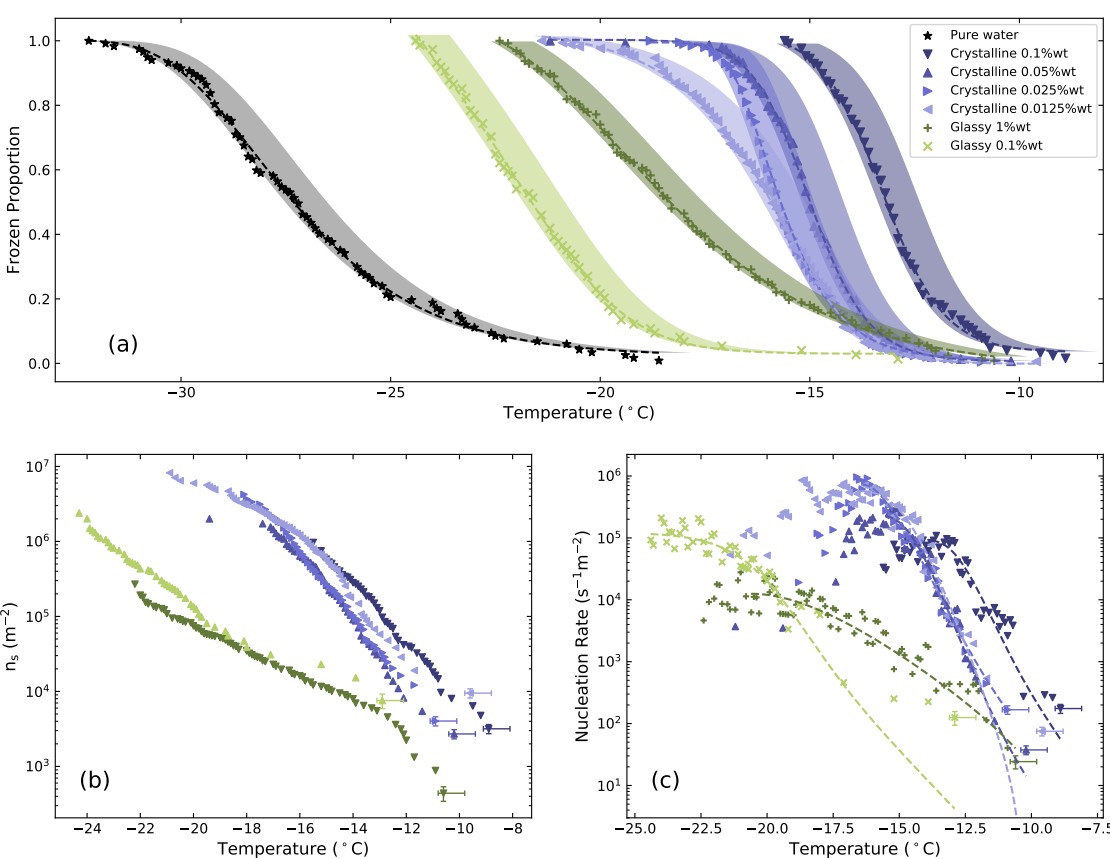

**Figure 5. a** Frozen fraction curves for 1 μl droplets of water containing different fractions of glassy and crystalline K-Feldspar. The background freezing rate of the instrument is also shown as the pure water frozen fraction. **b** Ice nucleation active site density, normalised by the surface area present in each droplet. **c** The nucleation rate calculated from classical nucleation theory. Only the error bars of the first data point of each sample are shown for clarity.