# Peer review of "A pyroelectric thermal sensor for automated ice nucleation detection"

_Atmospheric Measurement Techniques, 2019_

## Referee Comment (RC1) · Russell Perkins (Referee) · 5 Feb 2020

The manuscript describes proof-of-concept experiments using a pyroelectric sensor to detect ice nucleation events in a conventional cold plate instrument. This is, to my knowledge, the first time this type of sensor has been used in this type of instrument. Discussion of other instrumental techniques and their advantages and drawbacks is reasonable and complete. The experiments are brief but reasonable, allowing for comparison with data from different instruments and producing some novel data. The manuscript is publishable, but would be much improved if several specific comments below are addressed.

Specific Comments:

[Figure]

1) Lines 170-180, sample preparation: More details should be included on the process for making the glassy sample: temperatures used, quench process, milling size, any characterization of final material.

2) Lines 170-180: Feldspar materials can weather in aqueous solution, especially when acidic. This may be particularly important for ice nucleation, which depends on the structure of the mineral surface. A brief discussion of this is perhaps warranted, given that samples were stored suspended for up to a week. The glassy samples may also weather differently from the crystalline one. See refs:

Lee, M. R. & Parsons, I. Microtextural controls of weathering of perthitic alkali feldspars. Geochimica et Cosmochimica Acta 59, 4465–4488 (1995). https://doi.org/10.1016/0016-7037(95)00255-X

Lee, M. R., Hodson, M. E. & Parsons, I. The role of intragranular microtextures and microstructures in chemical and mechanical weathering: direct comparisons of experimentally and naturally weathered alkali feldspars. Geochimica et Cosmochimica Acta 62, 2771–2788 (1998). https://doi.org/10.1016/S0016-7037(98)00200-2

3) Paragraph around line 180: It doesn't appear that stochastic/binomial errors in n(T) were considered for the error estimates, which is the conventional way of doing this analysis and is almost always the dominant source of error. This should be remedied. A "score confidence interval" is the best approach at low numbers of freezing events. Justification for this, as well as equations for the calculation (eq. 2), can be found in:

Agresti, A. & Coull, B. A. Approximate Is Better than 'Exact' for Interval Estimation of Binomial Proportions. The American Statistician 52, 119–126 (1998). https://doi.org/10.2307/2685469

4) Due to the low mass fractions used in the solutions under study, effects noted in the following paper may become significant Although I believe good overlap in ns values (as shown in this manuscript) is an indication that corrections are not necessary:

Beydoun, H., Polen, M. & Sullivan, R. C. Effect of particle surface area on ice active site densities retrieved from droplet freezing spectra. Atmospheric Chemistry and Physics 16, 13359–13378 (2016). https://doi.org/10.5194/acp-16-13359-2016
* * *

---

## Referee Comment (RC2) · Anonymous Referee #3 · 13 Feb 2020

This paper describes the development of a novel method to perform ice nucleation experiments using a pyroelectric sensor, providing fast and cheap automated data collection. The important innovation in this paper is the use of the pyroelectric sensor, which provides an alternative to typical optical techniques used to identify freezing events in droplet freezing type experiments. This affords advantages such as cost-effective instrument design, and therefore this paper demonstrates and describes an innovative new instrument and is suitable for publication in AMT. I enjoyed reading and reviewing this paper and I think it will make a valuable addition to the literature. Before publication, I would recommend that the authors address the following:

General comments:

Whilst the experiments and analysis performed are sound, as a technique focused

paper it would be beneficial to demonstrate how broadly applicable this technique is. Could this, for instance, be used at different cooling rates that have been used in the literature such as 10 Kmin-1 (Broadley et al., 2012), on samples with steeper gradients in fraction frozen than feldspar (Pummer et al., 2015;Polen et al., 2016), on smaller droplets such as those generated by nano-printing or microfluidics (Peckhaus et al., 2016;Tarn et al., 2018)?

Specific comments / discussion:

Pg3 L73-74: "However, due to the Stefan-Boltzmann law infrared thermometry at low temperatures is limited to large droplets." I agree with this point but wonder if a limit be calculated as to the size of droplet / temperature range accessible? This is an interesting point, and this type of calculation or some sort of limiting value would add useful context if possible.

Pg5 L159: What would be the smallest droplet measurable by this technique? How sensitive are the films to the latent heat released? Do you anticipate a signal / noise issue at smaller droplet sizes?

Pg6 L174: Please expand on this method. What was the temperature that the sample was heated to? For how long? How was it rapidly quenched? How was it milled? As it stands, it would not be possible to repeat this process based on the information given.

Pg6 L179: Feldspar is susceptible to acid ageing. Did you measure the pH to ensure that the acid was removed by the Milli-Q rinses? How much rinsing was done?

Pg6 L183-185: "The values were 5.0 ± 0.7 m2g-1 and 1.8 ± 0.4 m2g-1 for crystalline and glassy K-feldspar respectively. The percentage errors associated with the surface area per unit mass dominate the error in calculating surface area present in each droplet, but are comparable to other experiments." I think this statement needs to be evidenced with citations. In some articles, the uncertainties in surface area per drop are not reported separately, so it is difficult to assess their magnitude. This is particularly true when the Poisson error is said to be the dominant source of uncertainty. The evidence you provide here demonstrates that this is important to account for, particularly when considering the sensitivity of ice nucleation measurements to surface area uncertainties (Alpert and Knopf, 2016).

Pg6 L185: The Poisson error is mentioned in the SI, but it is not propagated into the uncertainties (which instead use the surface area uncertainties). Can the authors comment on the calculation that led to the "exceeds 10%" comment in the SI and why these errors are not included?

Pg6 L185: Should this list also include weighing uncertainties?

Pg 6 L165: If the nucleant has a much steeper gradient in fraction frozen (for example pollen, fungal or bacterial INPs), would there be an overlap of signal, and what would be the limit of the experiment? In other words, how many droplets freezing per second could this method distinguish. This limit may also be interesting to determine for potential application if it were to be used for many smaller droplets, for example in a microfluidics experiment.

Pg6 L193: Did you perform experiments at cooling rates other than 1 °C/min? What would be the uncertainty based on literature thermal conductivity for different cooling rates?

Pg7 L196: Given that the background showed some heterogeneous nucleation, did you consider accounting for the background from your data using the differential spectra? (Vali, 2019)

Pg7 L208: Since you describe site specific nucleation, is it more appropriate to use freezing rate rather than nucleation rate as discussed by Vali? (Vali, 2014)

Pg7 L222: "The value of jhet(T) found for glassy crystalline K-feldspar here represents the average particle." Can you please clarify what is meant by the average particle here? Would the heterogeneity of the sample not bias the jhet(T) measurement? (Her-

[Figure]

bert et al., 2014;Holden et al., 2019)

Figure 3: It is made clear in the figure caption that positive spikes represent freezing events. What is the source of the negative spikes? If this is an artefact, is the same artefact possible with positive values (i.e. recording false positive signal)?

Figure 5: Whilst I agree that not displaying all error bars helps with clarity, I think it would be helpful to add more than just the first data points for ease of interpretation (perhaps at 25%, 50%, 75% and 100%?).

Figure 5b: As the sample used is the same feldspar as Atkinson (2013), perhaps it would be helpful to display the parameterisation from this paper, so that the performance of the new stage can be compared to that of the optical cold stage they used.

Figure 5b: Can the authors comment on the variation in ns for different wt% suspensions? In particular, there seems to be an offset between 0.5 wt% / 0.25 wt% and 0.1 wt% / 0.0125 wt% for crystalline feldspar. For example, at ns = 104 the uncertainties displayed do not explain the differences in the data. Is this expected based on the uncertainties in ns? Or could this relate to the length of time suspensions were kept for before experiments?

Technical Corrections:

SI: Is [64] a reference? If so, please correct to AMT format.

General comment: The spaces between sentences and references are inconsistent (sometimes there is a space and sometimes there isn't).

References

Alpert, P. A., and Knopf, D. A.: Analysis of isothermal and cooling-rate-dependent immersion freezing by a unifying stochastic ice nucleation model, Atmos. Chem. Phys., 16, 2083-2107, 10.5194/acp-16-2083-2016, 2016.

Broadley, S. L., Murray, B. J., Herbert, R. J., Atkinson, J. D., Dobbie, S., Malkin, T.

L., Condliffe, E., and Neve, L.: Immersion mode heterogeneous ice nucleation by an illite rich powder representative of atmospheric mineral dust, Atmos. Chem. Phys., 12, 287-307, 10.5194/acp-12-287-2012, 2012.

Herbert, R. J., Murray, B. J., Whale, T. F., Dobbie, S. J., and Atkinson, J. D.: Representing time-dependent freezing behaviour in immersion mode ice nucleation, Atmos. Chem. Phys., 14, 8501-8520, 10.5194/acp-14-8501-2014, 2014.

Holden, M. A., Whale, T. F., Tarn, M. D., O'Sullivan, D., Walshaw, R. D., Murray, B. J., Meldrum, F. C., and Christenson, H. K.: High-speed imaging of ice nucleation in water proves the existence of active sites, Science Advances, 5, eaav4316, 10.1126/sciadv.aav4316, 2019.

Peckhaus, A., Kiselev, A., Hiron, T., Ebert, M., and Leisner, T.: A comparative study of K-rich and Na/Ca-rich feldspar ice-nucleating particles in a nanoliter droplet freezing assay, Atmos. Chem. Phys., 16, 11477-11496, 10.5194/acp-16-11477-2016, 2016.

Polen, M., Lawlis, E., and Sullivan, R. C.: The unstable ice nucleation properties of Snomax$^{®}$ bacterial particles, Journal of Geophysical Research: Atmospheres, 121, 11,666-611,678, 10.1002/2016jd025251, 2016.

Pummer, B. G., Budke, C., Augustin-Bauditz, S., Niedermeier, D., Felgitsch, L., Kampf, C. J., Huber, R. G., Liedl, K. R., Loerting, T., Moschen, T., Schauperl, M., Tollinger, M., Morris, C. E., Wex, H., Grothe, H., Pöschl, U., Koop, T., and Fröhlich-Nowoisky, J.: Ice nucleation by water-soluble macromolecules, Atmos. Chem. Phys., 15, 4077-4091, 10.5194/acp-15-4077-2015, 2015.

Tarn, M. D., Sikora, S. N. F., Porter, G. C. E., O'Sullivan, D., Adams, M., Whale, T. F., Harrison, A. D., Vergara-Temprado, J., Wilson, T. W., Shim, J.-u., and Murray, B. J.: The study of atmospheric ice-nucleating particles via microfluidically generated droplets, Microfluidics and Nanofluidics, 22, 52, 10.1007/s10404-018-2069-x, 2018.

Vali, G.: Interpretation of freezing nucleation experiments: singular and stochastic;

sites and surfaces, Atmos Chem Phys, 14, 5271-5294, 2014.

Vali, G.: Revisiting the differential freezing nucleus spectra derived from drop-freezing experiments: methods of calculation, applications, and confidence limits, Atmos. Meas. Tech., 12, 1219-1231, 10.5194/amt-12-1219-2019, 2019.

---

## Referee Comment (RC3) · Anonymous Referee #2 · 17 Feb 2020

This manuscript by Cook et al. describes a new method for monitoring the freezing temperatures of microlitre water droplets using a pyroelectric polymer film sensor. The approach described is, to the best of my knowledge, new and would certainly be useful to the many researchers using these sorts of assays. The manuscript is very well written and presented and should be published. I have a few minor suggestions which the authors may wish to consider. These mostly align with the changes recommended in the two referee comments already posted.

Minor comments: The most important change to my mind would be some discussion of the likely limitations of the approach. This would be helpful for anyone looking to adapt pyroelectric sensors to other droplet freezing instruments or sample types. Specifically, some estimate of minimum droplet size and minimum supercooling (or combinations

thereof) that could be detected would be of interest. For instance, could this instrument reliably pick out freezing of microlitre droplets at around -2°C as can be induced by Snomax? (Wex et al., 2015) In this scenario the temperature change on freezing would be less than in the datasets presented and the freezing events would occur at much smaller time intervals.

My view is that the rates presented are better described as freezing rates rather than nucleation rates (Vali et al., 2015). Also, the nucleation rate is not derived from classical nucleation theory as stated in the caption to Fig. 5. Classical nucleation theory could be used to describe the nucleation rate but is not needed for calculating rates from experimental data.

As mentioned by the other referees a little more discussion of the nature of the samples and of the preparation of the glass samples would be of useful.

Vali, G., DeMott, P. J., Möhler, O., and Whale, T. F.: Technical Note: A proposal for ice nucleation terminology, Atmos. Chem. Phys., 15, 10263-10270, 10.5194/acp-15-10263-2015, 2015. Wex, H., Augustin-Bauditz, S., Boose, Y., Budke, C., Curtius, J., Diehl, K., Dreyer, A., Frank, F., Hartmann, S., Hiranuma, N., Jantsch, E., Kanji, Z. A., Kiselev, A., Koop, T., Möhler, O., Niedermeier, D., Nillius, B., Rösch, M., Rose, D., Schmidt, C., Steinke, I., and Stratmann, F.: Intercomparing different devices for the investigation of ice nucleating particles using Snomax[®] as test substance, Atmos. Chem. Phys., 15, 1463-1485, 10.5194/acp-15-1463-2015, 2015.

---

## Author Comment (AC1) · 21 Apr 2020

We attach our reply as a supplement. As our supplementary information has been largely re-written it has been included.

Please also note the supplement to this comment:
https://www.atmos-meas-tech-discuss.net/amt-2019-401/amt-2019-401-AC1-supplement.zip

---

## Author Response (AR1)

Author's Response

We would like to thank the referees for their useful comments and suggestions. Here we list the changes made in response to the referees' comments, as listed point by point in the individual replies. Red text indicates changes made, blue existing text for context.

Responses to referees

In response to anonymous referee 3 regarding detecting latent heat via infrared thermometry (line 73):

However, due to the Stefan-Boltzmann law infrared thermometry at low temperatures is usually limited to large droplets (Harrison 2018) although the latent heat released by droplets as small as 0.1 µL freezing has been reported (Kunert 2018).

Harrison, A. D. et al., (2018), Atmos. Meas. Tech, 11, 5629–5641. https://doi.org/10.5194/amt-11-5629-2018

Kunert, A. T. et al., (2018), Atmos. Meas. Tech, 11, 6327–6337. https://doi.org/10.5194/amt-11-6327-2018

In response to anonymous referees 2 and 3 regarding the temporal resolution of the thermal sensor (line 142):

The output from the charge amplifier was monitored using an analogue to digital converter (NI USB-6002), sampled at 1 kHz, which is fast enough to detect all droplets freezing, without creating unnecessarily large data files. For INPs that freeze over a very narrow temperature range, the sampling rate for this analogue to digital converter could be increased to 50 kHz to reduce the chance of near simultaneous freezes not being detected as separate events.

In response to anonymous referees 2 and 3 regarding the minimum droplet volume which could be detected the following was added (line 170):

The spatial variation in pyroelectric coefficient also means that droplets smaller than 1 µl could be detected in places. However, in order to guarantee detection across the whole surface the minimum size was set at 1 µl. The minimum droplet size detectable is also dependent on the minimum supercooling: assuming the droplet temperature returns to 0°C before freezing completely, the lower the supercooling, the lower will be the absolute temperature change on freezing and hence the lower the voltage pulse detected by the pyroelectric foil.

All three referees requested more information on the sample preparation (line 172):

The crystalline K-feldspar comes from the Bureau of Analysed Samples (BCS-CRM No. 376/1), as used by Atkinson et al. 2013. No further processing of this sample was done. The sample was crushed in a ball mill with agate balls before being sieved using a fine mesh (aperture size 20 µm).

The glassy K-feldspar sample was made from the crystalline sample melted in a platinum crucible. It was held at 1250°C overnight to remove moisture, before being heated to 1600°C for two hours. After this, the sample was removed from the furnace and allowed to quench in air. A few sections of the glass formed were examined under a polarizing microscope and no birefringent regions were observed. The glassy sample was then crushed and sieved using the same method described for the crystalline sample.

Referees 1 and 3 raised the issue of Feldspars again in suspension. This has been considered at the end of the sample preparation section:

Feldspar materials are susceptible to surface changes in aqueous solutions (Lee and Parsons, 1995) and when exposed to extreme pH (Kumar et al., 2018), which could lead to a change in their ice nucleating ability. Peckhaus et al. (2016) measured a 2°C decrease in freezing temperatures of K-feldspar stored in aqueous solution for five months. However, Kumar et al. (2018) recorded no change in the ice nucleating ability of crystalline K-feldspar after one week in water suspension and Harrison et al. (2016) noted no significant changes in freezing temperatures of crystalline K-feldspar due to time spent in water suspension. We assume that any aging of K-feldspar in aqueous solution is sufficiently slow to not have an effect on our results. Due to the identical chemical composition of the glassy sample we assume that any aging effects are similarly slow.

We have corrected a mistake concerning the comparison of errors on surface area to other literature as pointed out by referee 3. We apologize, as it was intended to compare the surface areas themselves, corrected as (line 202):

The surface area of both samples was measured via Brunauer-Emmett-Teller (BET) nitrogen gas absorption. Three repeats were taken, with the mean to extreme range used as the error. The values were 5.0 ± 0.7 $m^2g^{-1}$ and 1.8±0.4 $m^2g^{-1}$ for crystalline and glassy K-feldspar respectively, which are comparable to other experiments. The percentage errors associated with the surface area per unit mass dominate the error in calculating surface area present in each droplet, but are comparable to other experiments.

Queries on the errors contributing to the surface area present in each droplet from referees 1 and 3 have been addressed as follows (line 205):
There are also errors associated with the masses of K-feldspar and water when making suspensions, the volume of each droplet pipetted, and amount of material which settled out of suspension during pipetting (Tarn et al. 2018). These are particularly important for small droplet volumes and low concentrations (Beydoun et al. 2016, Knopf et al. 2020), however due to the relatively large droplet volumes used here they are insignificant compared to surface area per mass error.

Knopf, D. A. et al., (2020), NPJ Clim. Atmos. Sci. , 2, https://doi.org/10.1038/s41612-020-0106-4

Referee 3 asked if background freezing had been considered. We have added a section to the SI detailing how they were evaluated, and the following in the main text (line 203):

The influence of background freezing events on the liquid proportion curve of each experiment was calculated (more details in the supplementary information), but in all cases the corrected curve lay within the temperature errors.

In response to referee suggestions to evaluate the stochastic/binomial error in the liquid proportion data the analysis was redone to include them. Figure 5 has been replaced with the updated values and the following paragraph was added to the SI:

Stochastic errors were estimated using the Wilson score confidence interval on each temperature bin with more than one nucleation event. An example is shown in Figure S1. The errors in the liquid proportion were calculated based on how the minimum and maximum number of freezing events would affect the liquid proportion at that temperature bin, assuming the mean number of events were seen in all higher temperature bins. These errors were then combined with the errors in INP area in the calculation of $n_s$ and $J$ using standard propagation techniques.

Referees 2 and 3 brought up the clarification between freezing rate (Vali 2014) and nucleation rate. The following change was made to address this (line 250):

The value of $j_{het}$ found for glassy and crystalline K-feldspar here represents the freezing rate (Vali 2014) divided by the surface area measured by BET the average particle.

General Changes

Since the paper was submitted we have started using a new method to fit liquid proportion curves. This is detailed in the SI, with the following changes made in the text (line 225):

The dashed lines are generalised logistic functions (see supplementary information). These have no basis in theory, but provide good, monotonically decreasing, lines of best fit which can be differentiated analytically. The solid lines are taken from a fit assuming the liquid proportion curves follow a non-homogeneous Poisson process, referring to the fact that the rate constant is changing as a function of temperature. A full derivation of the fit can be found in the supplementary information.

In the new analysis Figure 5A was changed to liquid proportions from frozen fractions. This has led to changes in the text.

[revised manuscript text omitted]